# Consumption of Sericin Enhances the Bioavailability and Metabolic Efficacy of Chromium Picolinate in Rats

**DOI:** 10.3390/ijms262311505

**Published:** 2025-11-27

**Authors:** Chainarong Tocharus, Jiraphan Saelim, Manote Sutheerawattananonda

**Affiliations:** 1Department of Anatomy, Faculty of Medicine, Chiang Mai University, Chiang Mai 50200, Thailand; chainarongt@hotmail.com; 2School of Food Technology, Institute of Agricultural Technology, Suranaree University of Technology, Nakhon Ratchasima 30000, Thailand; jiraphan1238@gmail.com

**Keywords:** sericin, chromium picolinate, lipid metabolism, glucose regulation, adiposity, hypolipidemic effect

## Abstract

The effects of silk sericin (SS) supplementation on the functionality of chromium picolinate (CrPic) and lipid metabolism were assessed in male Sprague–Dawley rats to determine whether SS improves the bioavailability of CrPic and contributes to beneficial changes in lipid profiles and cardiovascular risk markers. Rats were administered different doses of SS (0, 1, 10, and 100 mg kg^−1^ body weight (BW)) in conjunction with CrPic (300 µg kg^−1^ BW) for 8 weeks through oral gavage. Body and organ weights, lipid profiles, glucose levels, chromium (Cr) accumulation, total protein, and adipocyte size were evaluated. Additionally, FTIR analysis was conducted to investigate the binding and release behavior of SS with CrPic. Although body weight and daily feed intake were comparable among groups, a significant increase in pancreas weight and reduction in omentum weight were observed across all CrPic-SS groups. Increased dosages of SS resulted in a significant reduction in triglyceride and plasma glucose levels. All CrPic and CrPic-SS treatments reduced LDL and total cholesterol while increasing HDL. Cr accumulation was elevated in the liver and kidneys of groups administered 10 and 100 mg kg^−1^ BW of SS, accompanied by a significant increase in total protein levels and a reduction in adipocyte size to less than 50 µm in all rats. FTIR analysis indicated that SS binds to CrPic at pH 2.0 and releases it at pH 7.0, demonstrating pH-dependent delivery similar to the gastrointestinal tract and possibly improved CrPic functionality. These findings indicate that SS improves the bioavailability of CrPic and positively affects lipid metabolism and cardiovascular risk markers in vivo.

## 1. Introduction

Overweight and obesity are global pandemics, especially in middle- and high-income countries [1]. The World Health Organization (WHO) has predicted that the rate of obesity and overweight will increase globally, impacting millions of people [1]. Obesity is the leading cause of glucose intolerance and diabetic onset symptoms [2]. The lifetime risk of type 2 diabetes (T2DM) at age 18 increased from 7.6% to 70.3% in underweight to very obese men, and from 12.2% to 74.4% in women. At age 65, the risk difference for overweight to very obese individuals compared to those of normal weight rose from 3.7% to 23.9% in men and from 8.7% to 26.7% in women [3]. T2DM can also lead to atherosclerosis and hyperlipidemia subsequent to the permanent onset of hyperglycemia [4]. Hyperlipidemia increases the risk of cardiovascular diseases (CVDs) and strokes, which can lead to serious consequences, including immobility [5,6]. Research has focused on addressing weight gain and obesity while maintaining muscle mass, given that obesity is the primary cause of T2DM and CVDs [7]. In addition to regular exercise, one medical intervention involves incorporating specific micronutrients, particularly metallic elements, into food intake [8]. Based on several studies, Cr has shown promising results in handling body weight and T2DM [9,10,11]. 

Cr (III), an essential component of the glucose tolerance factor (GTF), can help reduce hyperglycemia and hyperlipidemia in individuals with T2DM [12,13]. Oral administration of Cr (III) has been shown to mitigate symptoms resembling diabetes [13]. Nevertheless, the use of additional Cr (III) remains contentious for diabetes patients due to its absorption efficiency in the gastrointestinal tract dependent upon the chemical form of Cr present in the body [14]. In comparison to inorganic Cr (III) that can be converted into harmful Cr (VI) through oxidation [15,16], organic Cr (III) supplements enhance insulin efficacy and rectify glycolipid metabolism issues while exhibiting reduced cytotoxicity and genotoxicity [17]. Organic Cr (III) supplements, such as CrPic, Cr-propionate, Cr-histidinate, and Cr-nicotinate, have been shown to be useful. The recommended daily intake of CrPic is 50–200 μg, but excessive consumption may lead to negative consequences [18]. CrPic may enhance muscle insulin sensitivity and may be effective as a treatment for T2DM [19]. In vitro research has demonstrated that picolinate ligand-induced coordination of Cr enhanced the toxicity of CrPic to cultured cells, leading to a rise in apoptosis [20]. This suggests that while CrPic can have beneficial effects, its increased toxicity in certain conditions may pose risks, particularly in high doses. Thus, it is important to carefully evaluate using CrPic supplements, particularly for people who have underlying medical issues [21].

To overcome the poor absorption of Cr (III), one promising strategy involves chelation with natural biopolymers. While much of this research has focused on polysaccharides, which can form complexes with Cr (III) to augment hypoglycemic and hypolipidemic bioactivities [13,17,22,23]; however, proteins represent another major class of biopolymers with significant potential for mineral chelation [24]. Among these, silk sericin is a particularly compelling candidate. As a protein byproduct of the silk industry, sericin is not only a sustainable resource but is also exceptionally rich in amino acids with functional groups ideal for metal ion binding [24].

SS is a water-soluble globular protein with a molecular weight ranging from 20 to 310 kDa [25]. It covers the fibroin fiber layer and comprises around 20–30% of the cocoon’s total weight [26]. SS consists of 18 amino acids characterized by strongly polar side groups, containing hydroxyl, carboxyl, and amino groups [25], with aspartic acid and serine concentrations reaching 19% and 32%, respectively [27]. In addition to being safe for consumption, SS had no adverse effects in an investigation of its toxicity in pregnant rats and their fetuses [28]. This suggests that it is not only safe for general dietary use but also poses no risks during pregnancy. The hydroxyl and carboxyl groups on SS can bind or attract certain minerals through chelation [24]. Researchers discovered that foods with SS increased the absorption of iron, zinc, magnesium, and calcium by 41%, 41%, 21%, and 17%, respectively [29]. These elements were still excreted normally in the urine. This indicates that SS enhances the body’s ability to absorb essential minerals without causing any harmful side effects. Consequently, it supports overall health and nutrition while maintaining normal bodily functions. Previous studies have also demonstrated that SS and its hydrolysates derived from silk cocoons possess a wide range of biological activities, including hypoglycemic, antihypertensive, antioxidant, cholesterol-lowering, and colon cancer-preventive effects [30,31,32,33,34]. Interestingly, it has been demonstrated that even smaller molecules, such SS-derived oligopeptides, maintain a high degree of bioactivity, especially when it comes to lowering blood pressure [35], and blood sugar [36]. These results demonstrate the multifunctionality of SS, which helps prevent and treat metabolic and chronic disorders in addition to improving mineral absorption [33].

Prior studies indicate that silk SS beads effectively absorb hazardous Cr (VI) ions from industrial wastewater, achieving a maximum capacity of 33.76 mg g^−3^ at pH 2.0, along with a desorption rate of 73.19% under an alkaline condition [37]. Simple coagulation methods were implemented to produce SS and lignin composite beads, which exhibited superior efficacy in the removal of Cr (VI) in comparison to silk SS beads, obtaining a removal rate of over 90% and recycling 82% of it [38]. The crosslinking of chitosan with SS significantly enhanced the adsorption capacity of the chitosan/SS conjugate (CS) for Cr (VI) and methyl orange dye from aqueous solutions, achieving values of 139 mg g^−3^ and 385 mg g^−3^, respectively, while demonstrating effective regeneration [39]. CS is capable of reducing Cr (VI) to the less hazardous Cr (III). A maximal bio-absorption capacity of 0.25 mmol g^−3^ at 30 °C was attained by SS and alginate particles, which effectively removed Cr (III) and Cr (VI) from aqueous solutions [40]. This novel approach highlights SS’s potential as a biopolymer for Cr ion removal in wastewater treatment and suggests its applicability in enhancing Cr (III) bioavailability and efficacy through dietary supplementation in animal models. Limited research has documented the possible health advantages of SS when combined with CrPic in in vivo investigations. Taking advantage of SS’s capacity for Cr (III) adsorption and desorption at pH levels of 2.0 and 7.0, corresponding to the pH of a rat’s gastric and small intestine, we hypothesized that SS supplementation could increase the functionality of CrPic. Since Cr (III), in chromium picolinate is also a metal cation, it is chemically plausible that it would bind to these same functional groups. Our hypothesis was a logical extension of these prior findings that if sericin can effectively chelate and improve the bioavailability of other essential minerals, it should be able to do the same for Cr. The study examined how the treatment influenced body and organ weight, visceral fat content, adipocyte cell size, clinical chemistry parameters, and Cr accumulation in the internal organs of rats after 8 weeks of treatment. This study aimed to assess the long-term physiological effects of SS-enhanced CrPic, potentially clarifying its benefits for bioavailability and health outcomes.

## 2. Results

### 2.1. Characteristics of SS

Figure 1 illustrates the distinctive bands corresponding to the molecular weight of the SS protein extracted at 121 °C for durations of 60, 90, and 120 min, using pre-stained SDS-PAGE broad range standards as a reference marker. The broad nature of the bands is characteristic of the heterogeneous sericin protein mixture [41,42]. The SS protein extracted at 60 min had a molecular weight ranging from 132 to 76 kDa. After 90 and 120 min of treatment, the molecular weight of the SS protein ranged from 118 to 65 kDa and 113 to 61 kDa, respectively. The 60 min extraction was found to be optimal for obtaining SS with a higher molecular weight range suggesting a more intact structure. Table 1 shows the amino acid composition of SS consisting of 18 amino acids, with serine being the most abundant, followed by aspartic acid, glycine, threonine, and tyrosine consistent with the findings of Wu et al. (2008) [43].

### 2.2. SS-Cr Chelating Ability at Stomach and Intestinal pH Levels

The FTIR spectra of CrPic under simulated gastric digestion for 6 h (Figure 2) revealed distinct absorption bands. Additionally, absorption bands observed at 1050–1200 cm^−1^ were attributed to the C–O stretching of carboxylate groups, while those in the range of 500–700 cm^−1^ indicated metal–ligand vibrations [44]. The spectrum of CrPic alone exhibited distinct characteristic absorption peaks corresponding to functional groups of picolinic acid, such as C=O, C–N, and aromatic ring vibrations, confirming its spectral fingerprint [44]. In contrast, SS alone displayed no pronounced peaks in the regions associated with CrPic, consistent with its proteinaceous nature.

For the FTIR spectra of the CrPic–SS mixture after adjusting to simulated digestion pH, spectral changes were dependent on both incubation time and pH conditions. The black line represents CrPic alone, while the red line corresponds to SS alone. The blue, green, and purple lines represent the CrPic–SS complex at 0 h (pH 2.0), 2 h (pH 2.0), and 6 h (2 h at pH 2.0 followed by 4 h at pH 7.0), respectively. At 0 h and pH 2.0, the FTIR spectrum of the CrPic–SS mixture (blue line) exhibits characteristic peaks of both SS and CrPic, though the intensity of CrPic-specific peaks (1200–1000 cm^−1^) is notably lower than that of free CrPic (600–400 cm^−1^). After 2 h at pH 2.0 (green line), the complex maintains a similar spectral fingerprint, indicating that the CrPic–SS complex remains stable under gastric conditions. Upon adjustment to pH 7.0 and further incubation for 4 h (total of 6 h, purple line), the FTIR spectrum shows that while SS peaks are largely retained, the CrPic-specific peaks increase in intensity (1200–1000 cm^−1^). This indicates the partial release of CrPic from the SS complex under intestinal pH conditions. Kwak et al. (2013) [37] reported that Cr (VI) may be released when pH increases to 6.0–7.0, suggesting that at pH 7.0 after 4 h of digestive condition, partial release of CrPic from the complex may occur depending on the extent of digestion and the stability of the complex formed.

### 2.3. Effects on Body and Organ Weight

At the end of the experiment, there was no statistical difference in the daily amount of feed intake between groups. This suggests that SS appears to have no impact on the feed consumption of the rats. Furthermore, there was no statistical difference in the final body weight between groups (Table 2). However, all treatment groups tended to have decreased body weight when compared to those of the control group.

The intake of a standard diet that includes Cr did not influence feeding efficiency, consistent with the earlier study conducted by Mooney and Cromwell (1995) [45]. The final weight and food intake rates indicated minimal difference in growth rates between the control and treatment groups. This corroborates Hasten et al. (1997) [46], who concluded that Cr did not affect the growth rates of rats. It is consistent with the findings of Amoikon et al. (1995) [47] and Boleman et al. (1995) [48], who demonstrated that CrPic had no effect on growth rate and feed efficiency in pigs. Mooney & Cromwell (1995) [45] discovered that CrPic-augmented weight gain without affecting feed intake in pigs. However, this contradicts Page et al. (1993) [49], who initially suggested that Cr at doses of 50 and 200 ${\mu g\;kg}^{-1}$ could improve growth rates in pigs.

The omentum weight exhibited significant differences among the groups given CrPic together with all doses of SS at 1, 10, and 100 ${mg\;kg}^{-1}$ BW, compared to the control group (Table 3). The liver weight of the group received CrPic combined with SS at a dose of 100 ${mg\;kg}^{-1}$ BW decreased significantly compared to the other groups. All treatment groups had significant gains in pancreatic weight compared to the control group, with the CrPic and 100 ${mg\;kg}^{-1}$ BW SS group displaying the highest increase in pancreas weight. The weights of the seminal vesicle and epididymis decreased, while the weight of the testis increased; however, none of these differences were significant when compared to the control group and remained within the normal range [50,51].

All treatment groups exhibited a decrease in omentum weight. The rats treated with CrPic across all SS dosages showed a significant decrease in omentum weight. The concurrent administration of SS and CrPic resulted in a significant reduction in abdominal fat mass.

All groups showed an increase in pancreatic weight when compared to the control group. This is likely because the pancreas, which is both an exocrine and an endocrine gland, makes more insulin when CrPic or CrPic coupled with SS is given [52]. The liver weight decreased when CrPic was administered concurrently with SS at a dose of 100 ${mg\;kg}^{-1}$ BW.

### 2.4. Clinical Chemistry Analysis

Table 4 presents blood chemistry analysis for all rat groups. Plasma glucose levels significantly lowered in just two groups that received CrPic in combination with SS at doses of 10 and 100 ${mg\;kg}^{-1}$ BW compared to the control group. BUN exhibited no impact across all treatments. All treatment groups showed a reduction in cholesterol; however, those administered CrPic and the combination of CrPic and SS at all doses showed significant differences. Notably, the group receiving CrPic along with SS at a dose of 100 ${mg\;kg}^{-1}$ BW observed the most substantial cholesterol reduction, reaching 25.6%. All treatment groups exhibited an increase in HDL levels; however, the groups given CrPic in conjunction with all doses of SS had notable changes, which observed the most substantial increase around 22%. The groups administered CrPic and CrPic with all doses of SS had a significant reduction in LDL, with the group receiving CrPic and SS at a dose of 100 ${mg\;kg}^{-1}$ BW demonstrating the greatest percentage drop at 62.6%. All treatment groups exhibited a reduction in triglycerides, especially those given CrPic and SS across all doses. The cohort with CrPic and SS at doses of 10 and 100 ${mg\;kg}^{-1}$ BW had the most significant decrease percentage, at 50%. Total protein levels significantly increased in all experimental groups compared to the control group. CrPic and SS had similar effects in total protein levels.

Clinical chemistry results indicated that groups given CrPic together with SS at doses of 10 and 100 ${mg\;kg}^{-1}$ BW exhibited a significant reduction in blood glucose levels, in accordance with the findings of Zha et al. (2007) [53], which demonstrated that CrPic influenced insulin functionality in the metabolism of carbohydrates and lipids in animals [54]. Total protein also significantly increased across all treatment groups.

Table 5 shows the increased Cr accumulation in the kidneys and liver across all treatment groups, with significant differences observed within the groups given CrPic and SS at doses of 10 and 100 ${mg\;kg}^{-1}$ BW, with levels of 132.18 and 123.47 ${ng\;g}^{-1}$ in the kidneys and 114.93 and 116.30 ${ng\;g}^{-1}$ in the livers, respectively. The Cr levels in the groups administered CrPic with SS increased considerably at doses as low as 1 ${mg\;kg}^{-1}$ and significantly at 10 ${mg\;kg}^{-1}$ BW and 100 ${mg\;kg}^{-1}$ BW.

### 2.5. Effects on Adipocytes Size

Table 6 presents the dimensions of adipocytes in the omentum across all rat groups, using a diameter of 50 µm as a reference point. All six rats in the control group had adipocyte sizes more than 50 µm in diameter, representing 100% of the population. The treatment groups exhibited an increased tendency for a higher proportion of adipose cells measuring less than 50 µm in diameter, with significant differences found in the groups administered CrPic and CrPic in conjunction with all doses of SS. The group administered only CrPic was able to reduce adipocyte size to below 50 µm in diameter in up to 50% of the population. The combination of SS and CrPic resulted in an increase in adipose cells measuring smaller than 50 µm in diameter at 83% in the group administered SS at a dose of 1 ${mg\;kg}^{-1}$ BW and 100% at the groups fed SS at doses of 10, and 100 ${mg\;kg}^{-1}$ BW. The omentum-to-final body weight ratio for all groups (Table 2 and Table 3) exhibited a gradual decline from 0.18% in the control group to 0.146% in the group administered CrPic and a 100 ${mg\;kg}^{-1}$ BW dose of SS, with no significant differences in final body weight relative to the control group.

Figure 3 illustrates the physical characteristics and features of the adipose cells obtained from the omentum of all rat groups. At 40× magnification, the adipose cells exhibited irregular shapes, densely clustered with thin outer membranes and a translucent white center in each cell. The normal appearance of adipose cells in the control group, as shown in Figure 3A, resembles that of the group given SS at a dose of 100 ${mg\;kg}^{-1}$ BW (Figure 3B). Despite the comparable appearances of these two groups, their adipose cell sizes exhibited a slight but insignificant difference; the SS-fed group had 83% of adipose cells more than 50 μm, whereas the control group had 100% of cells exceeding this size (Table 6). Figure 3C,D displayed a similar morphology, characterized by small adipose cells densely aggregated in certain regions, intermittently interspersed with compact little blood vessels (capillaries). In the comparison of adipose cell images among all rat groups, those administered CrPic and SS at doses of 10 and 100 ${mg\;kg}^{-1}$ BW (Figure 3E,F) displayed smaller, densely packed adipose cells. Additionally, there was a uniform appearance of small blood vessels, with adipose cell sizes consistently measuring less than 50 μm across all rats.

## 3. Discussion

In the extraction process of sericin, the longer extraction times (90 and 120 min) may result in partial degradation of SS into smaller hydrolysates, potentially compromising its chelating properties. According to FTIR analysis, when SS and CrPic are mixed, the SS structure is left intact and CrPic forms a stable complex upon contact. This binding is likely mediated by amino acid residues of SS (e.g., serine, aspartic acid, and threonine) that provide potential coordination sites for Cr [29,55]. These observations are consistent with the report by Kwak et al. (2013) [37], which indicated that Cr (VI) exhibits a higher tendency to bind ligands at acidic pH 2.0; however, the complex was not yet fully formed. The retention of the SS characteristic peaks alongside the subdued CrPic peaks supports the notion that SS protects CrPic from early release in the acidic gastric environment [29]. Overall, this pH-responsive release behavior suggests that SS may act as a protective carrier, binding Cr in the stomach and facilitating its controlled release in the intestine, where CrPic absorption is more favorable. It binds Cr immediately under acidic conditions (pH 2.0), maintains the integrity of the complex during gastric transit, and releases Cr under neutral pH conditions (pH 7.0), simulating intestinal absorption [56]. SS may enhance mineral absorption by chelation through hydroxyl and carboxyl groups, maintaining minerals in a soluble state in the intestine. It would likely act by forming soluble Cr–SS complexes (especially Cr^3+^), preventing precipitation and enhancing intestinal transport [39]. The use of SS supplements to enhance the bioavailability of Cr may be advantageous, given that the general population ingests minimal quantities of Cr [57], coupled with the poor absorbability of Cr in the diet [58], resulting in a more prevalent and widespread Cr insufficiency.

At the conclusion of the study, a tendency for weight reduction was observed in rats administered CrPic together with SS, but without statistical significance. This may be due to the treatment’s two-month length being inadequate. A prolonged study may be required to assess its effect on weight reduction. The observed decrease in the weight of the treatment groups may be attributed to the improved Cr absorption mechanism in the rats, as previous research suggests that CrPic affects weight loss and promotes muscle growth and development in both humans and animals [59,60,61]. Due to various controversial findings, a conclusive determination of the influence of CrPic on growth rates and feed efficiency may still be insufficient. This experiment showed that SS might help the body absorb CrPic without changing the rate of growth or the efficiency of the feed. In contrast, it often promotes weight loss, maybe due to its role in enhancing the body’s absorption of CrPic, which improves the metabolism of carbohydrates and fats [54]. In addition, the effective absorption of Cr (III) from the CrPic-SS mixture into the bloodstream after passing through the digestive tract may be responsible for the decrease in omentum weight. Previous studies have shown that SS exhibits strong affinity for metal ions under acidic conditions due to its protonated amino groups, with reduced binding at higher pH levels that favors release [24]. Once delivered, CrPic has been shown to improve metabolic functions, in pigs, supplementation reduced fatty acid concentrations in adipose tissue [62,63]. In rodents, CrPic increased Cr accumulation in liver and brain and improved lipid and glucose homeostasis under high-fat-diet conditions [64,65].

An increase in pancreas weight can be a consequence of cellular growth and heightened enzyme production. Pancreas secretes enzymes like lipase for fat digestion in the duodenum [66], prompted by the enlargement of pancreatic cells that have been activated to perform additional functions. This leads to an increase in both the size of the cells and the production of pancreatic enzymes, ultimately resulting in an increase in pancreas weight [67]. The reduction in fat deposits in the liver directly contributes to a decrease in its overall weight. This may be due to higher rate of fat metabolism in the liver, leading to a lower amount of fat deposited in the liver and directly causing the overall weight reduction [68].

Cr may enhance insulin efficacy by facilitating the transfer of glucose and amino acids into muscle cells for energy metabolism and by stimulating ribosomal activity [69,70]. The lack of substantial changes in BUN levels indicates preserved kidney function without apparent impairment. Furthermore, reductions in cholesterol, LDL, and triglycerides were more significant at higher SS doses, which were similar to the findings of Okazaki et al. (2010) [71]. The risk of cardiovascular diseases decreased as the cardiovascular risk ratio (CRR) decreased. CRR, typically expressed as the ratio of total cholesterol to HDL cholesterol, is a reliable predictor of cardiovascular disease [72]. A CRR value of < 3 is generally considered safe, whereas values > 3 are associated with increased cardiovascular disease risk (Agu et al., 2024; Bhardwaj et al., 2013) [72,73]. In our study, all groups receiving SS along with CrPic demonstrated CRR values < 3, within the safe range, and lower than the control group. Furthermore, with a clear trend of decreasing CRR as SS supplementation increased, CVD risk could be reduced as indicated by the decreased CRR to 40% (from 2.40 in the control group to 1.46 in the CrPic 300 μg kg^−1^ BW + SS 100 mg kg^−1^ BW group). SS may lower CRR by reducing total cholesterol and LDL. Its antioxidant activities may also decrease the oxidative modification of LDL [74,75]. At the same time, the elevation of HDL facilitates reverse cholesterol transport, promoting the efflux of cholesterol from peripheral tissues and arterial walls back to the liver for excretion [76]. CrPic alone has been shown to improve lipid metabolism by enhancing insulin sensitivity [77], which indirectly reduces circulating triglycerides and LDL levels [10]. The combined supplementation of SS and CrPic may exert synergistic effects, with SS’s antioxidant and cholesterol-regulating activities complementing Cr’s role in glucose and lipid homeostasis [77,78]. Collectively, these mechanisms reduce lipid accumulation, thereby lowering CRR and ultimately decreasing the risk of cardiovascular diseases such as atherosclerosis, coronary artery disease, and stroke [72].

Total protein significantly increased across all treatment groups, suggesting an enhancement in protein metabolism presumably attributable to enhanced Cr absorption in internal organs, which subsequently induced changes in carbs and lipids metabolism, thereby facilitating muscle development. Our findings differed with the work of Zha et al. (2007) [53], which reported no significant effect of Cr supplementation in the form of CrCl_3_, CrPic, or CrNano on serum total protein levels in rats. This may be attributed to differences in Cr delivery and study design. Whereas Zha et al. administered Cr directly for 6 weeks using conventional or nano-formulations, our study employed SS as a pH-responsive carrier that possibly binds Cr at pH 2.0 and releases it at pH 7.0 like the acidic gastric pH and intestinal conditions [37], respectively, thereby enhancing intestinal absorption and tissue deposition. This improved bioavailability was reflected in higher Cr accumulation in the liver and kidney in the SS-supplemented groups (Table 5), which likely stimulated protein metabolism sufficiently to elevate circulating total protein. Moreover, the longer supplementation period in the present study (8 weeks versus 6 weeks) may have provided adequate time for these effects to manifest. Furthermore, since the researchers did not use SS combined with Cr in their investigation, it suggests that SS likely plays a crucial role in enhancing Cr absorption in the body compared to Cr alone.

The Cr concentration in the kidneys and liver also increased after the administration of either SS or CrPic separately. This suggests that SS has ability to increase the Cr absorbability from regular diet into both organs, similar to the effect observed when administering CrPic supplement to the animals, which enhanced Cr absorption into both organs via elevated blood concentrations. Moreover, high concentration of Cr was detected in the liver and kidneys as the concentration of Cr in feed increased, indicating that it is typically excreted from the body via feces and urine [79]. This suggests a substantial amount of Cr in the food that is available for absorption or increased absorption efficiency when supplemented with SS [29,80]. The observed dose-dependent increase in chromium accumulation in the liver and kidneys necessitates a thorough evaluation of potential long-term safety. Our own functional and biochemical data provide the first line of evidence for the safety of this intervention. Despite the significantly higher Cr deposition in the sericin-supplemented groups, we observed no corresponding signs of organ distress. Key markers of organ function remained stable across all groups; specifically, blood urea nitrogen (BUN) levels, a sensitive indicator of renal function, were unchanged, and there were no signs of hepatotoxicity, such as abnormal liver weight or visible lesions. This lack of toxicity, even with increased tissue accumulation, is strongly supported by extensive, long-term toxicological studies that have established a significant margin of safety for CrPic. A 20-week study by Anderson et al. (1997) [81] administered a diet containing 9 mg of chromium/kg BW per day, a dose approximately 30 times higher than that used in our study. While this led to substantial Cr accumulation in the liver (up to 500 ng/g) and kidney (up to 2000 ng/g), histological evaluation revealed no signs of toxicity. Furthermore, these findings are corroborated by a comprehensive 14-week study from the National Toxicology Program (2010), which used high doses of up to 4250 mg/kg. Even under this extreme exposure, which resulted in liver and kidney Cr concentrations of 1.3 µg/g and 7 µg/g, respectively, no significant adverse effects were reported. Therefore, we interpret the increased Cr deposition observed in our study not as a toxicological concern, but as a direct and positive indicator of sericin’s efficacy in enhancing the bioavailability and tissue uptake of Cr.

The treatments substantially reduced fat mass (FM) loss, whereas the reduction in lean body mass (LBM) loss was minimal, despite a trend toward lower body weight in the treatment groups. Significant differences in higher pancreatic weight, reduced blood glucose, and decreased levels of cholesterol, LDL, and triglycerides, in conjunction with increased HDL and total proteins (Table 4), indicate that the treatment groups exhibited superior metabolic function, particularly those administered all doses of CrPic and SS together. Previous research indicates that CrPic increases the bioavailability of Cr relative to other forms, hence enhancing insulin use and efficiency in humans as well as animal models [82]. Insulin affects body composition by facilitating the endogenous synthesis of fatty acids and triglycerides while also enhancing muscle protein synthesis [83]. CrPic has been thoroughly investigated for its unique ability to improve body composition, demonstrating efficacy in weight loss, calorie reduction, carbohydrate metabolism, and glucose metabolism in more than thirty-five human clinical studies [84]. SS has been reported to reduce blood glucose and cholesterol levels through its antioxidant properties. Previous studies have shown that SS and its derivatives upregulate the expression of key factors involved in the hepatic insulin PI3K/AKT signaling pathway, promote insulin production via pancreatic β-cell proliferation, decrease fat accumulation in hepatocytes, and improve both mitochondrial and endoplasmic reticulum integrity [30,35,85,86,87,88].

LBM loss often occurs together with weight loss and FM loss. This may result in adverse impacts on overall health. Maintaining LBM when consuming a considerable caloric deficit is challenging, regardless of the macronutrient distribution [84]. Tian et al. (2013) [89] reviewed and compared the previous research that illustrated several weight reduction techniques using common dietary supplements that have little effects on LBM loss or changes in body composition. In comparison to other weight loss supplements that minimize LBM loss, including fiber complex, green tea catechin, *Garcinia cambogia* blend, and *Irvingia gabonensis*, CrPic supplementation resulted in weight reduction accompanied by advantageous alterations in body composition, with 98% of the weight loss attributed to FM and merely 2% to LBM [89,90]. CrPic has shown efficacy as the optimal choice for enhancing body composition by preserving LBM. This study’s findings indicate that CrPic, when combined with SS at a minimal dosage of 1 ${mg\;kg}^{-1}$ BW, considerably enhances the efficiency of CrPic, which is recognized as the most effective dietary supplement for weight reduction with little loss of LBM. Various techniques have been used to chelate Cr, such as biochanin A [91] and the *Grifola frondosa* polysaccharide-Cr (III) complex, which enhanced the efficacy of Cr (III) supplementation [13]. Silk SS is an easy option when combined with CrPic for weight loss.

Previous studies have demonstrated that SS had the ability to absorb or desorb trace metal ions such as Mg, Fe, Zn, and Ca [29], including the toxic Cr (VI) ion [37], and both Cr (III) and Cr (VI) [39]. Comparable findings in other dietary proteins provide further support for this mechanism. For example, casein phosphopeptides have been shown to stabilize calcium and zinc ions through the formation of soluble complexes, thereby enhancing their intestinal absorption [92,93]. SS may function similarly to help solubilize and transport Cr across the intestinal epithelium due to the substantial amount of serine residues and polar side chains [94,95]. Moreover, its strong affinity toward multivalent ions suggests that SS may serve not only as a carrier that enhances mineral bioavailability but also as a modulator that influences the competitive uptake of essential and toxic metals [37,96].

There have been conflicting results from clinical trials assessing Cr supplementation in individuals with metabolic syndrome and type 2 diabetes, especially when it comes to lipid metabolism. For instance, supplementation with Cr in Indian patients with T2DM improved glycemic control but failed to produce significant improvements in lipid parameters [97]. Similarly, an 8-week randomized controlled trial administering 400 ${µg\;day}^{-1}$ CrPic in diabetic patients reported only modest reductions in total and LDL cholesterol, without significant effects on triglycerides or HDL levels [98]. Moreover, in obese nondiabetic adults with metabolic syndrome, even higher doses of CrPic (1000 ${µg\;day}^{-1}$ for 16 weeks) did not improve insulin sensitivity, serum lipids, or other cardiometabolic risk markers [99]. These restrictions suggest that Cr has a limited bioavailability and that better administration methods are required. Our study addresses this gap by demonstrating that silk SS, when co-administered with CrPic, enhances Cr bioavailability through pH-dependent release, resulting in significant reductions in triglycerides and plasma glucose, alongside consistent improvements in LDL, HDL, and total cholesterol levels. These findings not only corroborate the modest clinical benefits previously observed but also provide a mechanistic basis for superior lipid-modulating and cardioprotective effects, underscoring the translational potential of CrPic-SS formulations in metabolic disease management. The outcomes emphasize the value of integrating SS into dietary or therapeutic applications, paving the way for future research to validate its translational potential in human health.

## 4. Materials and Methods

### 4.1. Preparation of Silk SS and Its Characteristics

SS was extracted from raw silk yarn using distilled water (DW) in an autoclave at 121 °C for 60, 90, and 120 min [100]. The extract was dried using a double-drum dryer (New Way Manufacturing Co., Ltd., Samut Sakhon, Thailand) at 120 °C with a rotation speed of 85–95 rpm, then ground to a particle size of 0.075 mm in diameter. The resulting SS powder was stored in plastic bags at 25 °C in a desiccator.

### 4.2. Sericin Characterization

#### 4.2.1. Molecular Weight Determination

The molecular weight of SS was determined by SDS-PAGE following the protocol described by Weber and Osborn (1969) [101]. SS solution was mixed with sample buffer at a 1:2 ratio, boiled for 3–5 min, cooled, and loaded (10–15 µL) onto a 7% polyacrylamide gel using pre-stained broad-range protein standards Perfect Protein Markers (10–225 kDa) (M1) and ColorBurst™ Electrophoresis Marker (8–220 kDa) (M2) both were purchased from Sigma-Aldrich (Darmstadt, Germany), with sizes corresponding to those shown in Figure 1. Electrophoresis was performed at the current of 100 V for 40–56 min. Gels were stained with Coomassie Brilliant Blue R-250 (Bio-Rad Laboratories, Inc. (Hercules, CA, USA)) for 30–90 min and destained until clear protein bands were visible. Gel sheets were dried using cellophane and scanned with an Epson Perfection 4870 Photo scanner (model J131A, Seiko Epson Corp., Suwa, Nagano, Japan) to determine Retention Factor (Rf) values and estimate molecular weight. SS extracted at 60 min was selected in the study due to its broad molecular weight distribution (largest molecular weight range).

#### 4.2.2. Amino Acid Analysis

The SS powder from the 60 min autoclaving was sent to Central Laboratory Co., Ltd. (Bangkok, Thailand) for its amino acid compositions using the in-house method TE-CH-372 (based on Commission Directive 98/64/EC) [102]. The sample was hydrolyzed in 6 mol/L HCl (1:3, *w*/*v*) with 0.2 mL phenol at 110 °C for 22 h under vacuum. Amino acid analysis was performed using a Hitachi L8500A amino acid analyzer (Tokyo, Japan). For oral administration to the animals, SS solutions were freshly prepared daily by dissolving SS powder in distilled water (DW) to achieve the desired concentration.

### 4.3. FTIR Analysis of SS-Cr Chelating Ability at Stomach and Intestinal pH Levels

A solution was prepared by dissolving 0.10 g of CrPic and 0.15 g of SS in 20 mL of deionized water, with the pH adjusted to 2.0 using 1 M HCl to simulate gastric pH conditions (Šíma et al., 2019) [103]. The solution underwent microwave heating (800 W, 30 s) (CrPic + SS, 0 h pH 2.0) and was then incubated at 37 °C for 2 h. Following incubation, the solution was partitioned into two sets. The solution for Set 1 was shortly frozen at −80 °C (CrPic + SS, 2 h pH 2.0). In Set 2, the solution was adjusted to pH 7.0 using 1 M NaOH to simulate intestinal pH conditions (McConnell et al., 2008; Rodewald, 1976; Sciascia et al., 2016) [104,105,106] and subsequently incubated at 37 °C for 4 h before being frozen at −80 °C (CrPic + SS, 2 h at pH 2.0 + 4 h at pH 7.0). All frozen samples underwent freeze-drying for 48 h utilizing a freeze-dryer (Gamma 2-16 LSC, Christ, Germany) until a dry powder was achieved. The powders were kept in a desiccator at ambient temperature for further analysis.

Functional groups of the samples were analyzed using Fourier Transform Infrared (FTIR) spectroscopy with a Tensor 27 spectrometer (Bruker Optics, Ettlingen, Germany) equipped with a platinum Attenuated Total Reflectance (ATR) accessory adapted from Kwak et al. (2013) [37] and Manupa et al. (2023) [107]. A powdered sample (0.2–0.5 g) was placed directly onto the ATR crystal and pressed to ensure proper contact with the measurement surface. Spectra were recorded in the range of 4000–400 cm^−1^ at room temperature (25 ± 2 °C), with a resolution of 4 cm^−1^ and an accumulation of 64 scans.

### 4.4. Animals

Thirty-six male Sprague–Dawley rats (9 weeks old) were obtained from the National Bureau of Laboratory Animals, Mahidol University (Salaya, Nakhon Pathom, Thailand). They were housed under controlled environmental conditions with a 12:12 h light-dark cycle (light on from 6:00 a.m. to 6:00 p.m.) at a constant temperature of 25 ± 1 °C. The rats were provided with standard chow (Mouse Feed No. 082, C.P. Company, Bangkok, Thailand) and free access to reverse osmosis water throughout the experiment. All experimental procedures were approved by the Animal Ethics Committee of the Faculty of Medicine, Chiang Mai University, Thailand (Protocol No. 7/2555), and complied with the guidelines of the National Research Council of Thailand. Following each feeding test, the animals were euthanized via intraperitoneal injection of thiopental sodium at a lethal dosage of 50 ${mg\;kg}^{-1}$ BW. Blood samples were obtained immediately after cardiac puncture. Following this, the heart, spleen, thymus, liver, lungs, and kidneys were dissected under aseptic conditions, and their weights were documented [108].

### 4.5. SS Supplementation at Different Concentrations on Cr Functionality in Rats

This study used commercial GNC CrPic at a dosage of 200 mg (approximately 200 mg of Cr) per pill (Nutra Manufacturing Inc., Greenwich, NC, USA) as an organic source of Cr (III). The Cr content in each CrPic pill was preliminarily quantified using inductively coupled plasma mass spectrometry (ICP-MS), according to the method described by Anderson et al. (1996) [109] and an in-house protocol based on AOAC method 999.10 [110,111]. The rats were divided into six groups, with each group consisting of six rats. The sample size of six rats per group was selected based on previous metabolic studies from our laboratory [35,36,112] and the broader literature [113,114,115,116], which indicated that this number provides sufficient statistical power to detect significant effects for the primary biochemical endpoints, while adhering to the 3Rs principle for the ethical use of animals [117]. Group 1 served as the control group and received 1 mL of DW. Groups 2, 3, 4, 5, and 6 were administered tested substances mixed with 1 mL of DW. Group 2 received 100 ${mg\;kg}^{-1}$ BW of SS. Group 3 received a dose of 300 µg ${kg}^{-1}$ BW of CrPic. Group 4 was given a combination of 300 µg ${kg}^{-1}$ BW of CrPic and 1 ${mg\;kg}^{-1}$ BW of SS. Group 5 received 300 µg ${kg}^{-1}$ BW of CrPic and 10 ${mg\;kg}^{-1}$ BW of SS. Group 6 received 300 µg ${kg}^{-1}$ BW of CrPic and 100 ${mg\;kg}^{-1}$ BW of SS. All treatments were administered via oral gavage once daily in the morning, consistently between 8:00 and 9:00 AM for a duration of 8 weeks. The standard chow was provided ad libitum throughout the study. This timing was chosen to ensure a consistent administration protocol and to allow for absorption to begin without immediate interference from a large bolus of food.

The chromium picolinate dose of 300 µg ${kg}^{-1}$ BW was selected based on previous rodent studies demonstrating its metabolic efficacy [53,118,119]. This dose is also substantially below the No Observed Adverse Effect Level (NOAEL) established in comprehensive toxicological studies, confirming its safety [120].Throughout the experimental period, daily feed intake and BW were documented. Food consumption was calculated using the methods described by Zha et al. (2007) [53]. The rats were weighed before and after the treatment period to assess growth parameters.

### 4.6. Analysis of Blood Clinical Chemistry and Cr Levels in the Kidneys and Livers

At the end of the 8-week experimental period, rats were fasted overnight for 12 h to ensure a basal metabolic state. Following euthanasia, blood was collected immediately via cardiac puncture into tubes containing K_2_EDTA to prevent coagulation. Immediately following collection, the tubes were gently inverted 8–10 times to ensure thorough mixing of the blood with the anticoagulant and prevent microclot formation [121]. Subsequently, the samples were centrifuged at 3000× *g* for 15 min at 4 °C to separate the plasma. The resulting plasma supernatant was collected and promptly sent to the Med Star Lab (Chiang Mai, Thailand) for immediate analysis of all clinical chemistry parameters including plasma glucose, blood urea nitrogen (BUN), cholesterol, triglycerides (TGs), high-density lipoprotein (HDL), low-density lipoprotein (LDL), and total protein (TP). The livers and kidneys of the animals were excised and weighed. These two organs were selected because they are the established primary organs for Cr (III) accumulation, metabolism, and excretion. Existing evidence from comprehensive toxicological and confirms that Cr accumulation were found the highest in liver and kidney [120]. Subsequently, they went under analysis for Cr accumulation using ICP-MS equipment at the Chiang Mai Branch Central Laboratory Co., Ltd. (Chiang Mai, Thailand), which is accredited in accordance with ISO/IEC 17025 standards [122], following the protocols developed by Anderson et al. (1996) [109].

### 4.7. Weights of Internal Organs and Characteristics of Abdominal Adipose Tissue (Omentum)

The internal organs, including the omentum, were weighed to assess intergroup variations. Omental tissues were fixed in 10% formalin, sectioned into slices no thicker than 3 μm, and underwent hematoxylin and eosin (H&E) staining to evaluate histological changes according to the methodology outlined by Chavalittumrong et al. (2004) [108].

### 4.8. Data Analysis

All experimental data were expressed as mean ± standard deviation (SD). Statistical analyses were performed using one-way analysis of variance (ANOVA), followed by the Least Significant Difference (LSD) post hoc test to compare differences between groups at a statistically significant level of *p* < 0.05. Analyses were conducted using SPSS version 16.0 (SPSS Inc., Chicago, IL, USA).

## 5. Conclusions

The results of our study indicate that the combination of CrPic and varying doses of silk SS results in beneficial effects, such as improved lipid profiles, reduced blood glucose levels, decreased visceral fat accumulation, and smaller adipocyte size. These effects are particularly evident at SS doses of 10 and 100 ${mg\;kg}^{-1}$ BW. SS demonstrates a strong binding affinity for Cr (III) under acidic conditions that are comparable to the rat stomach environment. The binding is reversible, which facilitates the release of Cr ions in neutral conditions that are characteristic of the proximal small intestine. This process enhances the uptake of Cr into the bloodstream, thereby improving its bioavailability and distribution across various tissues. FTIR analysis confirmed that SS binds CrPic at pH 2.0 and releases it at pH 7.0, indicating a pH-dependent delivery mechanism in the gastrointestinal tract. In addition, SS, a protein byproduct of the silk reeling process, is sustainably organically produced within 25 days, offering an eco-friendly and value-added application. This approach has the potential to enhance the health outcomes of individuals at risk of metabolic disorders and to encourage the sharing of benefits among sericulture communities.

## Figures and Tables

**Figure 1 ijms-26-11505-f001:**
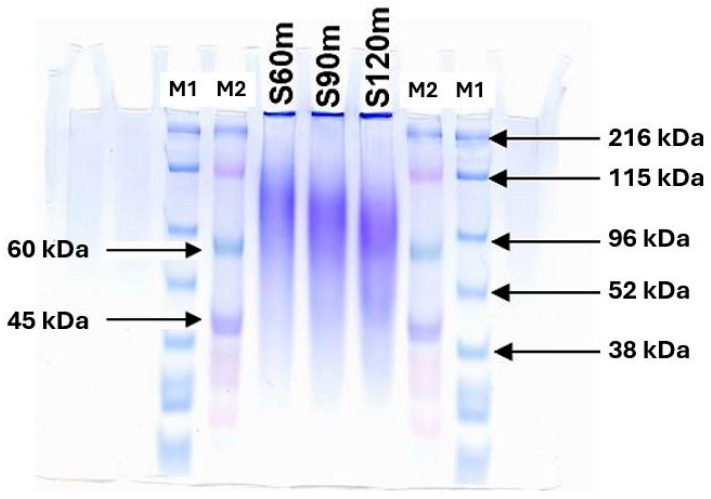
SDS-PAGE analysis of sericin protein extracted from raw silk yarn. Extraction at a temperature of 121 °C for durations of 60, 90, and 120 min is denoted by the symbols S60m, S90m, and S120m, respectively, with M1 and M2 representing the standard marker.

**Figure 2 ijms-26-11505-f002:**
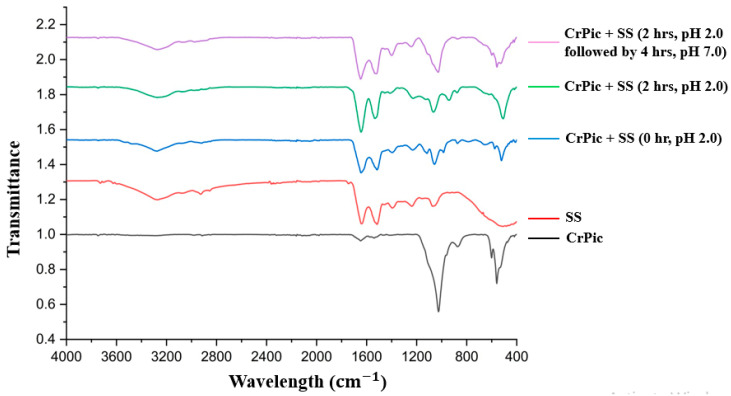
FTIR spectra of CrPic, SS, and their mixtures under simulated gastrointestinal conditions. The spectra include CrPic alone, SS alone, CrPic + SS at 0 h at pH 2.0, CrPic + SS after 2 h at pH 2.0, and CrPic + SS after 2 h at pH 2.0 followed by adjustment to pH 7.0 and incubation for an additional 4 h.

**Figure 3 ijms-26-11505-f003:**
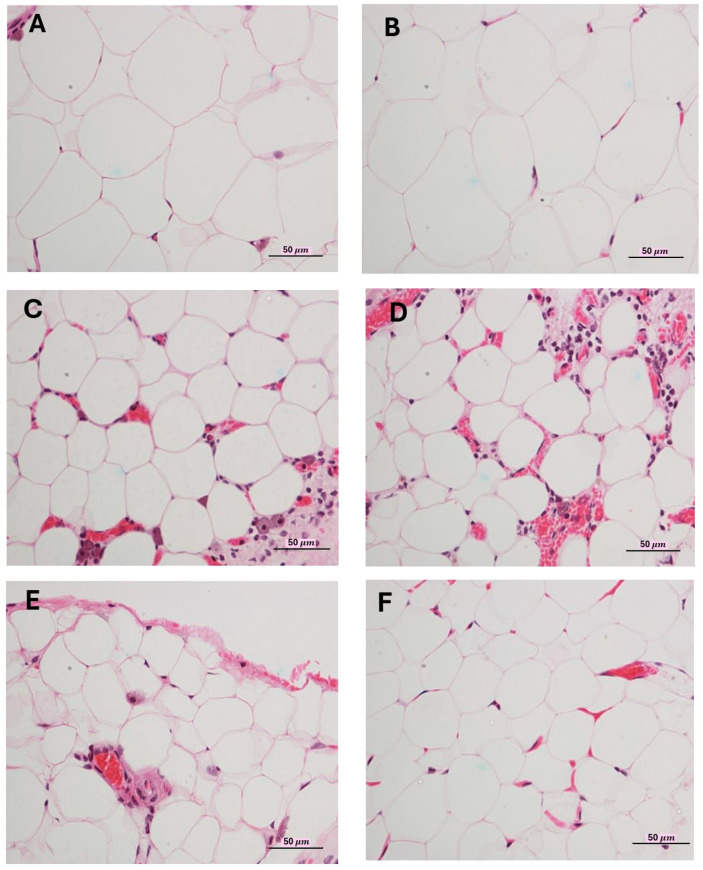
Microscopic images of adipose cells in the omentum of the control and treatment groups that received CrPic and CrPic with various doses of sericin at the end of an 8-week experimental period. (**A**): Control (40×), (**B**): SS 100 ${mg\;kg}^{-1}$ (40×), (**C**): CrPic 300 µg ${kg}^{-1}$ (40×), (**D**): CrPic 300 µg ${kg}^{-1}$ + SS 1 ${mg\;kg}^{-1}$ (40×), (**E**): CrPic 300 µg ${kg}^{-1}$ + SS 10 ${mg\;kg}^{-1}$ (40×), (**F**): CrPic 300 µg ${kg}^{-1}$ + SS 100 ${mg\;kg}^{-1}$ (40×).

**Table 1 ijms-26-11505-t001:** Amino acid compositions of sericin from raw silk yarn.

Amino Acids	**Concentration (mg 100 g^−1^)**	Limit of Detection (LOD) (mg)
Aspartic acid	18,373.10	100
Threonine	7726.64	100
Serine	32,472.87	100
Glutamic acid	5084.54	50
Glycine	8853.44	50
Alanine	3184.83	50
Cystine	<200.00	100
Valine	4345.81	50
Methionine	Not Detected	100
Isoleucine	691.44	50
Leucine	1200.34	50
Tyrosine	5402.71	100
Phenylalanine	462.93	100
Histidine	1877.87	50
Hydroxylysine	Not Detected	100
Lysine	3129.27	50
Arginine	4144.50	100
Hydroxyproline	Not Detected	200
Proline	556.41	100
Tryptophan	549.47	50

**Table 2 ijms-26-11505-t002:** Comparison of body weight and daily feed intake between control and treatment groups given CrPic and CrPic with various doses of SS during an 8-week experimental period.

Parameter	Control	$\mathrm{SS}\;100\;{mg\;kg}^{-1}$ BW	$\mathrm{CrPic}\;300\;{\mu g\;kg}^{-1}$ BW	$\mathrm{CrPic}\;300\;{\mu g\;kg}^{-1}$$\;\mathrm{BW}\;+\;\mathrm{SS}\;1\;{mg\;kg}^{-1}$ BW	$\mathrm{CrPic}\;300\;{\mu g\;kg}^{-1}$$\;\mathrm{BW}\;+\;\mathrm{SS}\;10\;{mg\;kg}^{-1}$ BW	$\mathrm{CrPic}\;300\;{\mu g\;kg}^{-1}$ BW $+\;\mathrm{SS}\;100\;{mg\;kg}^{-1}$ BW
Initial body weight (g)	330.00 ± 15.49	325.00 ± 20.74	341.67 ± 16.02	325.00 ± 30.82	355.00 ± 22.58	336.67 ± 18.62
Final body weight (g)	451.67 ± 18.35	431.67 ± 21.37	445.00 ± 24.29	423.33 ± 42.27	430.00 ± 55.14	430.00 ± 33.47
Daily feed intake (g per day)	19.76 ± 1.42	18.57 ± 0.93	19.52 ± 1.73	18.93 ± 1.42	18.81 ± 0.81	18.45 ± 0.75

Values are expressed as mean ± SD.

**Table 3 ijms-26-11505-t003:** Comparison of internal organ weights of the control and treatment groups given CrPic and CrPic with various doses of SS at the end of an 8-week experimental period.

Parameter	Control	SS $100\;{mg\;kg}^{-1}$ BW	CrPic $300\;{\mu g\;kg}^{-1}$ BW	$\mathrm{CrPic}\;300\;{\mu g\;kg}^{-1}$ BW $+\;\mathrm{SS}\;1\;{mg\;kg}^{-1}$ BW	$\mathrm{CrPic}\;300\;{\mu g\;kg}^{-1}$ BW $+\;\mathrm{SS}\;10\;{mg\;kg}^{-1}$ BW	$\mathrm{CrPic}\;300\;{\mu g\;kg}^{-1}$ BW $+\;\mathrm{SS}\;100\;{mg\;kg}^{-1}$ BW
Omentum (g)	0.81 ± 0.09	0.75 ± 0.12	0.74 ± 0.06	0.71 ± 0.08 *	0.70 ± 0.02 *	0.63 ± 0.04 *
Heart (g)	1.39 ± 0.06	1.46 ± 0.19	1.51 ± 0.07	1.54 ± 0.09	1.45 ± 0.21	1.39 ± 0.18
Lung (g)	2.10 ± 0.61	1.69 ± 0.34	2.50 ± 0.76	2.05 ± 0.62	2.49 ± 0.70	1.92 ± 0.58
Liver (g)	13.30 ± 0.36	12.89 ± 1.11	13.00 ± 1.00	11.87 ± 1.28	12.85 ± 2.06	11.39 ± 0.96 *
Kidney (g)	2.51 ± 0.18	2.40 ± 0.24	2.53 ± 0.19	2.36 ± 0.37	2.56 ± 0.26	2.42 ± 0.20
Adrenal gland (g)	0.07 ± 0.02	0.06 ± 0.03	0.08 ± 0.01	0.06 ± 0.02	0.06 ± 0.02	0.05 ± 0.01
Pancreas (g)	0.86 ± 0.17	1.08 ± 0.15 *	1.09 ± 0.26 *	1.10 ± 0.12 *	1.09 ± 0.14 *	1.26 ± 0.19 *
Spleen (g)	0.95 ± 0.06	0.90 ± 0.04	0.96 ± 0.10	0.91 ± 0.09	0.97 ± 0.07	0.94 ± 0.05
Prostate gland (g)	0.46 ± 0.07	0.49 ± 0.10	0.41 ± 0.11	0.44 ± 0.08	0.45 ± 0.07	0.40 ± 0.10
Seminal vesicle (g)	1.61 ± 0.15	1.57 ± 0.33	1.53 ± 0.37	1.43 ± 0.31	1.40 ± 0.27	1.42 ± 0.07
Epididymis (g)	1.02 ± 0.28	1.12 ± 0.09	1.17 ± 0.09	1.18 ± 0.28	1.20 ± 0.17	1.06 ± 0.23
Testis (g)	2.21 ± 0.74	2.57 ± 0.60	2.62 ± 0.18	2.63 ± 0.26	2.74 ± 0.26	2.75 ± 0.89

Values are expressed as mean ± SD, * *p* < 0.05 compared to control group; six rats per group.

**Table 4 ijms-26-11505-t004:** Comparison of clinical chemistry blood of the control and treatment groups given CrPic and CrPic with various doses of SS at the end of an 8-week experimental period.

Parameter	Control	SS $100\;{mg\;kg}^{-1}$ BW	CrPic $300\;{\mu g\;kg}^{-1}$ BW	$\mathrm{CrPic}\;300\;{\mu g\;kg}^{-1}$ BW $+\;\mathrm{SS}\;1\;{mg\;kg}^{-1}$ BW	$\mathrm{CrPic}\;300\;{\mu g\;kg}^{-1}$ BW $+\;\mathrm{SS}\;10\;{mg\;kg}^{-1}$ BW	$\mathrm{CrPic}\;300\;{\mu g\;kg}^{-1}$ BW $+\;\mathrm{SS}\;100\;{mg\;kg}^{-1}$ BW
$\mathrm{Glucose}\;({mg\;dL}^{-1})$	158.33 ± 17.06	155.33 ± 12.19	152.00 ± 8.00	152.50 ± 6.95	143.33 ± 6.00 *	139.83 ± 15.88 *
$\mathrm{BUN}\;{(mg\;dL}^{-1})$	23.33 ± 1.03	23.83 ± 1.17	21.83 ± 1.47	23.00 ± 1.55	23.33 ± 3.14	21.33 ± 3.67
$\mathrm{Cholesterol}\;{(mg\;dL}^{-1})$	91.17 ± 3.92	85.17 ± 4.17	80.00 ± 8.37 *	81.67 ± 9.33 *	75.33 ± 3.88 *	67.83 ± 8.16 *
$\mathrm{HDL}\;{(mg\;dL}^{-1})$	38.00 ± 7.07	42.17 ± 3.66	42.83 ± 2.71	45.83 ± 5.81 *	46.00 ± 2.83 *	46.33 ± 6.62 *
$\mathrm{LDL}\;{(mg\;dL}^{-1})$	35.17 ± 2.14	33.00 ± 2.28	18.17 ± 2.32 *	19.00 ± 8.44 *	15.00 ± 5.97 *	13.17 ± 1.72 *
$\mathrm{Triglyceride}\;{(mg\;dL}^{-1})$	98.33 ± 38.52	82.67 ± 16.82	88.83 ± 53.92	70.83 ± 10.98	49.00 ± 7.56 *	48.67 ± 12.39 *
$\mathrm{Total}\;\mathrm{protein}\;{(g\;dL}^{-1})$	5.63 ± 0.30	6.27 ± 0.16 *	6.27 ± 0.22 *	6.15 ± 0.29 *	6.50 ± 0.30 *	6.15 ± 0.39 *

Values are expressed as mean ± SD, * *p* < 0.05 compared to control group; six rats per group.

**Table 5 ijms-26-11505-t005:** Comparison of the amount of Cr accumulated in the liver and kidneys of the control and treatment groups given CrPic and CrPic with various doses of SS at the end of an 8-week experimental period.

Treatment	Kidney (${ng\;g}^{-1}$)	Liver (${ng\;g}^{-1}$)
**Control**	95.75 ± 3.87	98.18 ± 6.33
$\mathrm{SS}\;100\;{mg\;kg}^{-1}$ **BW**	103.72 ± 24.35	99.67 ± 12.34
$\mathrm{CrPic}\;300\;{\mu g\;kg}^{-1}$ **BW**	104.93 ± 13.73	110.61 ± 16.23
$\mathrm{CrPic}\;300\;{\mu g\;kg}^{-1}$**BW** $+\;\mathrm{SS}\;1\;{mg\;kg}^{-1}$**BW**	112.09 ± 21.65	107.88 ± 10.06
$\mathrm{CrPic}\;300\;{\mu g\;kg}^{-1}$**BW** $+\;\mathrm{SS}\;10\;{mg\;kg}^{-1}$**BW**	132.18 ± 20.77 *	114.93 ± 7.74 *
$\mathrm{CrPic}\;300\;{\mu g\;kg}^{-1}$**BW** $+\;\mathrm{SS}\;100\;{mg\;kg}^{-1}$**BW**	123.47 ± 13.06 *	116.30 ± 8.56 *

Values are expressed as mean ± SD, * *p* < 0.05 compared to control group; six rats per group.

**Table 6 ijms-26-11505-t006:** Comparison of the size of adipose cells in the omentum of the control and treatment groups given CrPic and CrPic with various doses of SS at the end of an 8-week experimental period.

Treatment	Size of Adipose Cell	
	<50 µm (%)	>50 µm (%)
**Control**	0	6/6 (100)
$\mathrm{SS}\;100\;{mg\;kg}^{-1}$ **BW**	1/6 (17)	5/6 (83)
$\mathrm{CrPic}\;300\;{\mu g\;kg}^{-1}$ **BW**	3/6 (50) *	3/6 (50) *
$\mathrm{CrPic}\;300\;{\mu g\;kg}^{-1}$ $\;\mathrm{BW}\;+\;\mathrm{SS}\;1\;{mg\;kg}^{-1}$ **BW**	5/6 (83) *	1/6 (17) *
$\mathrm{CrPic}\;300\;{\mu g\;kg}^{-1}$ $\;\mathrm{BW}\;+\;\mathrm{SS}\;10\;{mg\;kg}^{-1}$ **BW**	6/6 (100) *	0 *
$\mathrm{CrPic}\;300\;{\mu g\;kg}^{-1}$**BW** $+\;\mathrm{SS}\;100\;{mg\;kg}^{-1}$**BW**	6/6 (100) *	0 *

Values are expressed as mean ± SD, * *p* < 0.05 compared to control group; six rats per group.

## Data Availability

The original contributions presented in this study are included in the article. Further inquiries can be directed to the corresponding author.

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
