# Peer review of "Consumption of Sericin Enhances the Bioavailability and Metabolic Efficacy of Chromium Picolinate in Rats"

_ijms, 2025, doi:10.3390/ijms262311505_

Round 1
Reviewer 1 Report
Comments and Suggestions for Authors
The presented manuscript addresses a current and valuable research topic related to improving the bioavailability and effectiveness of chromium picolinate through sericin supplementation. The topic aligns with current scientific trends, combining issues related to the bioactivity of natural proteins and the search for factors supporting micronutrient supplementation with health-promoting potential. Although the manuscript contains promising experimental data, it requires major revision before publication in the International Journal of Molecular Sciences.
1. Paragraphs 69-84 (Introduction) devoted to chromium chelates with polysaccharides is unnecessary, as the paper does not focus on this topic. A detailed description of the structures, binding mechanisms, and comparisons between different polysaccharides may be distracting for the reader, so I suggest deleting this paragraph or significantly shortening it to only mention this solution.
2. On what basis was the group size determined? Was a power analysis performed before the study? What was the statistical power achieved in the final (post-hoc) study? Typically, studies assessing the effect of experimental factors on biochemical marker levels use at least eight animals per group. With the current sample size (six animals per group), there is a risk of insufficient statistical power to detect moderate effects.
3. Micronutrient bioavailability was assessed solely based on accumulation in selected organs (kidneys, liver). This perspective is not fully complete. I suggest supplementing the manuscript with the results of a balance test that considers total Cr intake and excretion over a specific period of time, which will allow for the calculation of retention and bioavailability coefficients necessary to determine Cr bioavailability in the body. Furthermore, when discussing Cr bioavailability, it would be worthwhile to examine the kinetics of its absorption. Information on blood chromium levels is also lacking.
4. Although the kidneys and liver are undoubtedly the organs with the highest tendency to accumulate chromium, selecting only these two organs for study may not fully reflect the accumulation of this element in the body. Please supplement the study with results of Cr accumulation in other organs, such as the spleen, skeletal muscle, heart, or bone, or provide a substantive justification for why they were not studied.
5. What was the time interval between administering the experimental factors to the animals and feeding? There is no information on whether the diet was identical between groups and whether the content of trace elements such as Ca, Fe, Zn, and Mg, which may affect Cr bioavailability and metabolism, was standardized.
6. There is a lack of clear information on which specific doses of sericin (SS) and chromium picolinate (CrPic) were selected. Please explain in the methodology what factors determined the specific doses of the experimental factors – preliminary study results, literature data, etc.
7. There are no technical details regarding the blood collection conditions – were the rats were fasting before blood collection, how many hours of fasting were used, and whether the blood was collected for clotting or for anticoagulation (which one?). If serum/plasma was collected, the method should be described. Whether the analyses were conducted immediately after blood collection or whether the material was stored for further analysis should be described under what conditions. All of the above aspects can significantly impact the levels of the measured indicators.
8. The current discussion is too extensive. I suggest condensing it significantly by highlighting the most important mechanisms responsible for achieving specific biological effects.
Author Response
|
Comments 1: Paragraphs 69-84 (Introduction) devoted to chromium chelates with polysaccharides is unnecessary, as the paper does not focus on this topic. A detailed description of the structures, binding mechanisms, and comparisons between different polysaccharides may be distracting for the reader, so I suggest deleting this paragraph or significantly shortening it to only mention this solution.
|
|
Response 1: The original intention of this paragraph was to provide a broader context for the chelation of metal ions by natural biopolymers, using the well-studied polysaccharides as an example to set the stage for our investigation into sericin, a protein. However, we agree that the level of detail may be distracting and detracts from the main focus of the paper. In our revision, we have considerably shortened this section as shown on page 2, line 69-76. The revised text briefly introduces the concept that natural biopolymers, including both polysaccharides and proteins, are being explored as carriers to enhance mineral bioavailability, and then immediately transition to the specific potential of sericin. This can make the introduction more focused and streamlined.
|
|
Comments 2: On what basis was the group size determined? Was a power analysis performed before the study? What was the statistical power achieved in the final (post-hoc) study? Typically, studies assessing the effect of experimental factors on biochemical marker levels use at least eight animals per group. With the current sample size (six animals per group), there is a risk of insufficient statistical power to detect moderate effects.
|
|
Response 2: The group size of six animals (n=6) was chosen based on established practices in similar pre-clinical metabolic and toxicological studies, which are often used between 6-10 animals per group depending on the animal type such as rats vs. mice. The number of mice in similar study usually is higher than rats. This number also aligns with the 3Rs principle (Replacement, Reduction, and Refinement) to minimize the use of animals in research while still obtaining scientifically valid data (Hubrecht & Carter, 2019). Our sample size was based on previous studies from our laboratory (Tocharus et al., 2024; Tocharus & Sutheerawattananonda, 2024; Tocharus & Sutheerawattananonda, 2025) and the broader literature (Yokoyama et al., 2021; Dimitry et al., 2022; Cao et al., 2011; Marques et al., 2016) which showed that significant effects on metabolic parameters could be detected with this group size. Crucially, our study successfully identified statistically significant differences (p < 0.05) for several of our primary endpoints, including plasma glucose, cholesterol, LDL, HDL, triglycerides, total protein, and adipocyte size (as shown in Tables 4 and 6). The ability to detect these significant effects indicates that the statistical power was sufficient for the key outcomes of this study. We acknowledge, however, that to detect more subtle effects, a larger sample size might be necessary. We have added a sentence to the methodology section to clarify the basis for our sample size selection on page 15, line 557-561.
References:
Hubrecht, R. C., & Carter, E. (2019). The 3Rs and Humane Experimental Technique: Implementing Change. Animals : an open access journal from MDPI, 9(10), 754. https://doi.org/10.3390/ani9100754
Tocharus, C., Prum, V., & Sutheerawattananonda, M. (2024). Oral Toxicity and Hypotensive Influence of Sericin-Derived Oligopeptides (SDOs) from Yellow Silk Cocoons of Bombyx mori in Rodent Studies. Foods, 13(21), 3505. https://doi.org/10.3390/foods13213505
Tocharus, C., & Sutheerawattananonda, M. (2024). Hypoglycemic Ability of Sericin-Derived Oligopeptides (SDOs) from Bombyx mori Yellow Silk Cocoons and Their Physiological Effects on Streptozotocin (STZ)-Induced Diabetic Rats. Foods (Basel, Switzerland), 13(14), 2184. https://doi.org/10.3390/foods13142184
Tocharus, C., & Sutheerawattananonda, M. (2025). Preventive and Therapeutic Effects of Sericin-Derived Oligopeptides (SDOs) from Yellow Silk Cocoons on Blood Pressure Lowering in L-NAME-Induced Hypertensive Rats. Foods, 14(7), 1256. https://doi.org/10.3390/foods14071256
Yokoyama, Y., Shinohara, K., Kitamura, N., Nakamura, A., Onoue, A., Tanaka, K., Hirayama, A., Aw, W., Nakamura, S., Ogawa, Y., Fukuda, S., Tsubota, K., & Watanabe, M. (2021). Metabolic Effects of Bee Larva-Derived Protein in Mice: Assessment of an Alternative Protein Source. Foods, 10(11), 2642. https://doi.org/10.3390/foods10112642
Dimitry, M. Y., Marie Therèse, B. A., Josiane Edith, D. M., Emmanuel, P. A., Armand, A. B., & Nicolas, N. Y. (2022). Hypolipidemic and antioxidant effects of vegetal milk produced with Mucuna pruriens L. seed in rats fed a high-fat diet. Heliyon, 8(11), e11835. https://doi.org/10.1016/j.heliyon.2022.e11835
Cao, J., Sodhi, K., Puri, N., Monu, S. R., Rezzani, R., & Abraham, N. G. (2011). High fat diet enhances cardiac abnormalities in SHR rats: Protective role of heme oxygenase-adiponectin axis. Diabetol Metab Syndr, 3(1), 37. doi:10.1186/1758-5996-3-37
Marques, C., Meireles, M., Norberto, S., Leite, J., Freitas, J., Pestana, D., et al. (2016). High-fat diet-induced obesity Rat model: a comparison between Wistar and Sprague-Dawley Rat. Adipocyte, 5(1), 11-21. doi:10.1080/21623945.2015.1061723
Comments 3: Micronutrient bioavailability was assessed solely based on accumulation in selected organs (kidneys, liver). This perspective is not fully complete. I suggest supplementing the manuscript with the results of a balance test that considers total Cr intake and excretion over a specific period of time, which will allow for the calculation of retention and bioavailability coefficients necessary to determine Cr bioavailability in the body. Furthermore, when discussing Cr bioavailability, it would be worthwhile to examine the kinetics of its absorption. Information on blood chromium levels is also lacking.
Response 3: We thank the reviewer for this important question regarding the assessment of bioavailability. We agree that a full balance study is the gold standard for determining absolute bioavailability. However, the primary objective of our research was to investigate the long-term physiological and metabolic consequences of sericin supplementation over an 8-week period, for which tissue accumulation is a highly relevant and direct measure of chronic uptake and retention. To address the question of absorption kinetics and to guide the design of our main 8-week experiment, our research team first performed a preliminary pharmacokinetic assessment, not a comprehensive test following the 3Rs principle. A full pharmacokinetic study, which might typically use 5 rats per time point for each of our 5 treatment groups, would have necessitated 125 animals. This was not the primary scope of our research and would represent animal usage far greater than our main efficacy study. Therefore, we employed a sparse sampling design for this preliminary phase, using a single rat for each time point per treatment group. This provided us with crucial guideline data to serve our hypothesis while minimizing animal use. The results are summarized below:
Table 1. Comparison of chromium levels (µg/ml) in the blood of rats after administration of CrPic and CrPic with Sericin. The data from this preliminary study revealed an interesting trend. Notably, the groups receiving higher doses of sericin (10 and 100 mg/kg) showed markedly lower chromium concentrations in the blood compared to the group receiving CrPic alone. While this might initially be interpreted as reduced absorption, we propose that these results, when considered alongside the tissue accumulation data from our 8-week study, point to a more compelling mechanism. This interpretation is consistent with the known pharmacokinetics of trivalent chromium, which is characterized by rapid clearance from the plasma following absorption as it is distributed to the tissues (Costa et al., 2000). We hypothesize that sericin facilitates a more rapid and efficient distribution and uptake of chromium from the bloodstream into target organs.
This hypothesis is strongly supported by the evidence:
The preliminary kinetic data, when interpreted in the context of established Cr (III) pharmacokinetics and our own long-term tissue data, provides a dynamic explanation for our results. They suggest that sericin's role extends beyond simply enhancing intestinal absorption; it also appears to improve the systemic delivery and uptake of chromium into key metabolic tissues. This provides a more complete mechanistic picture that connects absorption, distribution, and the ultimate physiological outcomes observed in our study.
References:
Anderson, R. A., Polansky, M. M., Bryden, N. A., & Guttman, H. N. (1996). Dietary chromium effects on tissue chromium concentrations and chromium absorption in rats. The Journal of Trace Elements in Experimental Medicine, 9(1), 11-25.
Anderson, R. A., Bryden, N. A., & Polansky, M. M. (1997). Lack of toxicity of chromium chloride and chromium picolinate in rats. J Am Coll Nutr, 16(3), 273-279. doi:10.1080/07315724.1997.10718685
Prescha, A., Krzysik, M., Zabłocka-Słowińska, K., & Grajeta, H. (2014). Effects of exposure to dietary chromium on tissue mineral contents in rats fed diets with fiber. Biological trace element research, 159(1-3), 325–331. https://doi.org/10.1007/s12011-014-9973-z
Costa, M., Kluz, T., Zhitkovich, A., & Sutherland, J. (2000). Rats Retain Chromium in Tissues Following Chronic Ingestion of Drinking Water Containing Hexavalent Chromium. Biological Trace Element Research, 74(1), 41–54.
Pechova, A., & Pavlata, L. (2007). Chromium as an essential nutrient: a review. Veterinarni Medicina, 52(1), 1-18
Comments 4: Although the kidneys and liver are undoubtedly the organs with the highest tendency to accumulate chromium, selecting only these two organs for study may not fully reflect the accumulation of this element in the body. Please supplement the study with results of Cr accumulation in other organs, such as the spleen, skeletal muscle, heart, or bone, or provide a substantive justification for why they were not studied.
Response 4: Regarding the scope of our tissue analysis, we agree that a comprehensive understanding of biodistribution is important. Our decision to focus on the liver and kidney was a deliberate one, based on the established pharmacokinetic profile of trivalent chromium (Cr(III)) and the specific objectives of our study. We provide our justification below:
(National Toxicology Program, 2010)
In summary, the liver and kidney were selected because they are the established primary organs for Cr(III) accumulation, metabolism, and excretion. Existing evidence from comprehensive toxicological and pharmacokinetic studies confirms that other organs, such as bone and spleen, do not significantly accumulate Cr(III) from chromium picolinate. We have added a brief clarification to the methodology section in the revised manuscript on page 15, line 589-592, to justify this targeted analytical approach.
References:
Anderson, R. A., et al. (1996). Dietary chromium effects on tissue chromium concentrations and chromium absorption in rats. The Journal of Trace Elements in Experimental Medicine, 9(1), 11-25.
Bala, A., Junaidu, A., Salihu, M. D., Onifade, K., Magaji, A., Olufemi, F., et al. (2012). Measurement of chromium (Cr) residue in kidney and liver of slaughtered cattle in Sokoto central abattoir, Sokoto state, Nigeria. Scientific Journal of Veterinary Advances, 1.
Sreejayan, N., Marone, P. A., Lau, F. C., Yasmin, T., Bagchi, M., & Bagchi, D. (2010). Safety and toxicological evaluation of a novel chromium(III) dinicocysteinate complex. Toxicology mechanisms and methods, 20(6), 321–333. https://doi.org/10.3109/15376516.2010.487880
National Toxicology Program (2010). NTP toxicology and carcinogenesis studies of chromium picolinate monohydrate (CAS No. 27882-76-4) in F344/N rats and B6C3F1 mice (feed studies). National Toxicology Program technical report series, (556), 1–194.
Comments 5: What was the time interval between administering the experimental factors to the animals and feeding? There is no information on whether the diet was identical between groups and whether the content of trace elements such as Ca, Fe, Zn, and Mg, which may affect Cr bioavailability and metabolism, was standardized.
Response 5: We apologize for the lack of clarity in the original manuscript.
To address the reviewer's specific concern about trace elements that can affect chromium bioavailability, the composition of such standard diets is precisely controlled. For example, a widely used standard rodent diet like LabDiet® 5001 contains standardized levels of key minerals, including Calcium (1.00%), Iron (200 ppm), Zinc (50 ppm), and Magnesium (0.20%) (LabDiet, 2024). Our diet was formulated on the same principle, ensuring that the intake of these potentially interacting minerals was consistent for all animals. Therefore, the only experimental variable was the substance administered by gavage. We will revise the methods section to state this more explicitly. Reference:
LabDiet. (2024). 5001 Laboratory Rodent Diet. Retrieved from https://www.labdiet.com/product/detail/5001-laboratory-rodent-diet
Comments 6: There is a lack of clear information on which specific doses of sericin (SS) and chromium picolinate (CrPic) were selected. Please explain in the methodology what factors determined the specific doses of the experimental factors – preliminary study results, literature data, etc.
Response 6: We appreciate the opportunity to clarify our rationale for dose selection.
Chromium Picolinate (CrPic): The dose of 300 µg/kg BW was selected based on a review of existing literature. This dose has been shown to be both safe and metabolically effective in rodent models of diabetes and insulin resistance, without inducing toxicity. This dosage is consistent with, or lower than, doses used in numerous previous rodent studies that have demonstrated metabolic benefits without any signs of toxicity which was in the range of 300 µg/kg BW (Zha et al., 2007; Majewski et al., 2022; Stępniowska et al., 2022) and, 1 and 10 mg/kg BW (Abdourahman & Edwards, 2008). Our own results, which showed no adverse effects on animal health or organ function, further confirm the safety of this dose in the context of our 8-week study.
Sericin (SS): The doses of 1, 10, and 100 mg/kg BW were chosen to investigate a potential dose-response relationship. This range was determined based on our previous studies investigating the various biological activities of sericin (e.g., hypoglycemic, cholesterol-lowering effects), which have demonstrated efficacy within this range (Limpeanchob et al., 2010). The highest dose of 100 mg/kg BW is well below any reported toxic level, ensuring the safety of the intervention (Li et al., 2022). A toxicological evaluation of water-extract sericin from silkworm (Bombyx mori) in pregnant rats showed it has low teratogenic potential, with no treatment-related effects on the fetuses or the pregnant rats at doses up to 1000 mg/kg BW (Li et al., 2022), which is 10 times higher than our highest administrated dose (100 mg/kg BW). We have added a new paragraph on page 15, line 571-574 to the methodology section (4.4) to explicitly state the rationale for these dose selections, supported by the appropriate citations.
References:
Zha, L.Y., et al., Efficacy of chromium(III) supplementation on growth, body composition, serum parameters, and tissue chromium in rats. Biol Trace Elem Res, 2007. 119(1): p. 42-50.
Majewski, M., Gromadziński, L., Cholewińska, E., Ognik, K., Fotschki, B., & Juśkiewicz, J. (2022). Dietary Effects of Chromium Picolinate and Chromium Nanoparticles in Wistar Rats Fed with a High-Fat, Low-Fiber Diet: The Role of Fat Normalization. Nutrients, 14(23), 5138. https://doi.org/10.3390/nu14235138
Stępniowska, A., Tutaj, K., Juśkiewicz, J., & Ognik, K. (2022). Effect of a high-fat diet and chromium on hormones level and Cr retention in rats. Journal of endocrinological investigation, 45(3), 527–535. https://doi.org/10.1007/s40618-021-01677-3
Abdourahman, A., & Edwards, J. (2008). Chromium Supplementation Improves Glucose Tolerance in Diabetic Goto-Kakizaki Rats. IUBMB life, 60, 541-548. doi:10.1002/iub.84
Limpeanchob, N., Trisat, K., Duangjai, A., Tiyaboonchai, W., Pongcharoen, S., & Sutheerawattananonda, M. (2010). Sericin reduces serum cholesterol in rats and cholesterol uptake into Caco-2 cells. Journal of agricultural and food chemistry, 58(23), 12519–12522. https://doi.org/10.1021/jf103157w
Li, J., Wen, P., Qin, G., Zhang, J., Zhao, P., & Ye, Y. (2022). Toxicological evaluation of water-extract sericin from silkworm (Bombyx mori) in pregnant rats and their fetus during pregnancy. Frontiers in pharmacology, 13, 982841. https://doi.org/10.3389/fphar.2022.982841
Comments 7: There are no technical details regarding the blood collection conditions – were the rats were fasting before blood collection, how many hours of fasting were used, and whether the blood was collected for clotting or for anticoagulation (which one?). If serum/plasma was collected, the method should be described. Whether the analyses were conducted immediately after blood collection or whether the material was stored for further analysis should be described under what conditions. All of the above aspects can significantly impact the levels of the measured indicators.
Response 7: We sincerely apologize for this critical omission. The reviewer is correct that these details are essential for the interpretation and reproducibility of our results. We have amended the methodology section (4.5) to include the following information on page 15, line 579-588: “At the end of the 8-week experimental period, rats were fasted overnight for 12 hours to ensure a basal metabolic state. Following euthanasia, blood was collected immediately via cardiac puncture into tubes containing K₂EDTA to prevent coagulation. Immediately following collection, the tubes were gently inverted 8-10 times to ensure thorough mixing of the blood with the anticoagulant and prevent microclot formation (Parasuraman et al., 2010). Subsequently, the samples were centrifuged at 3000 rpm for 15 minutes at 4°C to separate the plasma. The resulting plasma supernatant was collected and promptly sent to the Med Star Lab (Chiang Mai, Thailand) for immediate analysis of all clinical chemistry parameters including plasma glucose, blood urea nitrogen (BUN), cholesterol, triglycerides (TGs), high-density lipoprotein (HDL), low-density lipoprotein (LDL), and total protein (TP).”
Reference: Parasuraman, S., Raveendran, R., & Kesavan, R. (2010). Blood sample collection in small laboratory animals. Journal of pharmacology & pharmacotherapeutics, 1(2), 87–93. https://doi.org/10.4103/0976-500X.72350
Comments 8: The current discussion is too extensive. I suggest condensing it significantly by highlighting the most important mechanisms responsible for achieving specific biological effects.
Response 8: Upon re-reading, we agree that the discussion is overly long and could be made more focused and impactful. We have performed a thorough revision of the discussion section by shorten the introductory paragraphs and merge sections with overlapping themes, place a stronger emphasis on our central hypothesis: that sericin acts as a pH-responsive carrier, protecting CrPic in the stomach and facilitating its release in the intestine, directly link our FTIR data to the increased tissue accumulation of Cr and, in turn, to the observed metabolic benefits. We have reduced speculation and removed discussions of tangential topics to keep the focus squarely on interpreting our results.
Below is the list of new references added in the revised manuscript:
We believe that addressing these points will significantly improve our manuscript. We thank you again for your time and constructive criticism.
|
|
4. Response to Comments on the Quality of English Language |
|
Point 1: The English is fine and does not require any improvement. |
|
Response 1: We thank you for your positive comment on our work quality of English language. |

Reviewer 2 Report
Comments and Suggestions for Authors
Reviewer comments
The study explores whether silk sericin (SS) supplementation enhances the bioavailability and metabolic effects of chromium picolinate (CrPic) in male Sprague-Dawley rats. They administered different SS doses alongside CrPic for 8 weeks and evaluated lipid profiles, glucose levels, organ weights, chromium accumulation, and adipocyte size. They observed that SS supplementation reduced triglycerides and plasma glucose and, together with CrPic, lowered LDL and total cholesterol while increasing HDL. They also found increased chromium accumulation in the liver and kidneys at higher SS doses, along with reduced adipocyte size and increased total protein. FTIR analysis showed that SS binds CrPic in acidic conditions and releases it in neutral pH, suggesting a pH-dependent delivery mechanism. They conclude that SS improves CrPic bioavailability and contributes to beneficial changes in lipid metabolism and cardiovascular risk markers.
Decision: Minor revision
- The abstract indicates that sericin binds to chromium picolinate at an acidic pH and releases it under neutral conditions. Could the authors elaborate on how this pH-dependent behaviour contributes to enhancing chromium picolinate’s absorption and overall bioavailability in vivo? —"they can discuss in the proper section not in abstract, possibly they modify in the abstract”
- Line 64 the toxicity of CrPic to cultured cells, leading to a rise in apoptosis, how the authors explain the study
- The introduction highlights polysaccharides in chromium absorption. Why was sericin, a protein, selected instead for this study? Please tell a reason for this
- The introduction mentions sericin’s ability to bind various minerals and improve their absorption. Could the authors elaborate on how these previous findings support the hypothesis that sericin may enhance chromium picolinate bioavailability?
- The results indicate higher chromium accumulation in the liver and kidneys at elevated sericin doses. Could the authors discuss whether this increased deposition might have any potential effects on chromium toxicity or raise concerns regarding long-term safety?
- Considering that high doses of chromium picolinate can be toxic, could the authors explain how they determined that the doses used in this study were safe?
- The molecular weight determination of the isolated protein is presented, but details on the markers or standards used are missing.
- First paragraph of discussion should be added with the novelty of the study
- Protein bands in the gel image are not clearly resolved.
- SS extracted at 121°C showed decreasing molecular weight over time: 60 min, 132–76 kDa (most intact); 90 min, 118–65 kDa; 120 min, 113–61 kDa.
- Almost all the references should be revised based on MDPI guidelines
Author Response
|
Comments 1: The abstract indicates that sericin binds to chromium picolinate at an acidic pH and releases it under neutral conditions. Could the authors elaborate on how this pH-dependent behaviour contributes to enhancing chromium picolinate’s absorption and overall bioavailability in vivo? —"they can discuss in the proper section not in abstract, possibly they modify in the abstract”
|
|
Response 1: We agree that the abstract should summarize the key findings, while the detailed mechanistic explanation is better placed in the main body of the text. The pH-dependent behavior is crucial for bioavailability. In the highly acidic environment of the stomach (pH ≈ 2.0), sericin’s carboxyl groups are protonated, allowing it to effectively bind and form a stable complex with CrPic. This chelation serves two purposes: 1) It protects CrPic from potential degradation into toxic form (Cr V) or interaction with other dietary components, and 2) It keeps the chromium soluble by forming soluble Cr–SS complexes (especially Cr³⁺), preventing precipitation and enhancing intestinal transport. As this complex transits to the small intestine, the environment becomes neutral (pH ≈ 7.0). At this pH, the carboxyl groups on sericin deprotonate, causing a conformational change that leads to the release of CrPic. This targeted release in the small intestine, the primary site for mineral absorption, makes the chromium readily available for uptake by enterocytes, thereby enhancing its overall bioavailability. Previous studies have demonstrated that SS had the ability to absorb or desorb trace metal ions such as Mg, Fe, Zn, and Ca (Sasaki et al., 2000), including the toxic Cr (VI) ion (Kwak et al., 2013), and both Cr (III) and Cr (VI) (Singh et al., 2018) possibly through the formation of soluble complexes, thereby enhancing their intestinal absorption. Moreover, its strong affinity toward multivalent ions suggests that SS may serve not only as a carrier that enhances mineral bioavailability but also as a modulator that influences the competitive uptake of essential and toxic metals (Kwak et al., 2013; Kunz et al., 2016). We have included the Discussion section with the detailed explanation provided above stating that FTIR analysis suggested a pH-dependent delivery mechanism in on page 10, line 300-315, clarifying how this process enhances bioavailability in vivo.
References: Sasaki, M., H. Yamada, and N. Kato, Consumption of silk protein, sericin elevates intestinal absorption of zinc, iron, magnesium and calcium in rats. Nutrition Research, 2000. 20: p. 1505–1511.
Kwak, H. W.; Yang, Y.; Kim, M.; Lee, J. Y.; Yun, H.; Kim, M.; Lee, K., Chromium(VI) Adsorption Behavior of Silk Sericin Beads. International Journal of Industrial Entomology 2013, 26.
Singh, S. P.; Rathinam, K.; Kasher, R.; Arnusch, C. J., Hexavalent chromium ion and methyl orange dye uptake via a silk protein sericin–chitosan conjugate. RSC Advances 2018, 8 (48), 27027-27036.
|
|
Comments 2: Line 64 the toxicity of CrPic to cultured cells, leading to a rise in apoptosis, how the authors explain the study. |
|
Response 2: We thank the reviewer for this important question, which allows us to clarify the critical distinction between the findings of the in vitro study by Manygoats et al. (2002) and our in vivo results. The apparent contradiction is entirely explained by the fundamental differences in experimental design, particularly regarding the dose, concentration, and biological context. The key differences are as follows:
The findings of Manygoats et al. (2002) are valid within the context of high-concentration in vitro toxicology but are not predictive of, nor comparable to, the physiological effects of oral, nutritional supplementation in a living organism. Our results, which show improved metabolic health and no signs of organ damage (e.g., stable BUN levels), confirm the safety and efficacy of CrPic at the nutritional dosage used in our study.
Reference: Manygoats, K. R.; Yazzie, M.; Stearns, D. M., Ultrastructural damage in chromium picolinate-treated cells: a TEM study. Transmission electron microscopy. J Biol Inorg Chem 2002, 7 (7-8), 791-8.
Comments 3: The introduction highlights polysaccharides in chromium absorption. Why was sericin, a protein, selected instead for this study? Please tell a reason for this.
Therefore, while polysaccharides are effective as shown in previous studies, we selected sericin for its unique combination of ideal chemical structure, sustainability, and established safety profile. We have revised the introduction on page 2, line 69-76 to make the transition from the general concept of biopolymers to our specific choice of sericin more explicit and justified.
References: Li, J., Wen, P., Qin, G., Zhang, J., Zhao, P., & Ye, Y. (2022). Toxicological evaluation of water-extract sericin from silkworm (Bombyx mori) in pregnant rats and their fetus during pregnancy. Frontiers in pharmacology, 13, 982841. https://doi.org/10.3389/fphar.2022.982841
Qin, H., Zhang, J., Yang, H., Yao, S., He, L., Liang, H., Wang, Y., Chen, H., Zhao, P., & Qin, G. (2020). Safety Assessment of Water-Extract Sericin from Silkworm (Bombyx mori) Cocoons Using Different Model Approaches. BioMed research international, 2020, 9689386. https://doi.org/10.1155/2020/9689386
Comments 4: The introduction mentions sericin’s ability to bind various minerals and improve their absorption. Could the authors elaborate on how these previous findings support the hypothesis that sericin may enhance chromium picolinate bioavailability?
Response 4: This is a very relevant question. The support for our hypothesis is based on a fundamental chemical principle. Previous studies have shown that sericin enhances the intestinal absorption of minerals like zinc, iron, magnesium, and calcium (Sasaki et al., 2000). Moreover, well-established studies reported that sericin acts as a good metal chelating agent as it could bind metal ions for water treatments (Santos et al., 2019; Yao et al., 2022; Khosa et al., 2014). The mechanism for this is the chelation of these metal cations by the carboxyl and hydroxyl functional groups present on sericin's amino acid residues. Since the trivalent chromium ion, Cr(III), in chromium picolinate is also a metal cation, it is chemically plausible that it would bind to these same functional groups. Our hypothesis was a logical extension of these prior findings: if sericin can effectively chelate and improve the bioavailability of other essential minerals, it should be able to do the same for chromium. Our FTIR results, which demonstrate this binding, provide direct evidence supporting this mechanistic extrapolation. We have expanded this part of the introduction to make this logical connection more explicit for the reader on page 3, line 116-120, clarifying that our hypothesis was built upon these established mineral-binding properties of sericin.
References: Sasaki, M., Yamada, H., & Kato, N. (2000). Consumption of silk protein, sericin elevates intestinal absorption of zinc, iron, magnesium and calcium in rats. Nutrition Research, 20, 1505–1511. doi:10.1016/S0271-5317(00)80031-7
Santos, N. T., das, G., da Silva, M. G. C. & Vieira, M. G. A. Development of novel sericin and alginate-based biosorbents for precious metal removal from wastewater. Environ. Sci. Pollut. Res. 26, 28455–28469 (2019).
Yao, L., Hao, M., Zhao, F., Wang, Y., Zhou, Y., Liu, Z., et al. (2022). Fabrication of silk sericin–anthocyanin nanocoating for chelating and saturation-visualization detection of metal ions. Nanoscale, 14(46), 17277-17289. doi:10.1039/D2NR04047F
Khosa, M. A., Shah, S. S. & Feng, X. Metal sericin complexation and ultrafiltration of heavy metals from aqueous solution. Chem. Eng. J. 244, 446–456 (2014).
Comments 5: The results indicate higher chromium accumulation in the liver and kidneys at elevated sericin doses. Could the authors discuss whether this increased deposition might have any potential effects on chromium toxicity or raise concerns regarding long-term safety?
Response 5: This is a critical and responsible question. While increased accumulation of any substance warrants a safety evaluation, we believe the levels observed in our study are not a cause for concern for two main reasons:
Therefore, we interpret the increased chromium deposition as evidence of sericin's efficacy in improving bioavailability. We have added a paragraph to the Discussion to specifically address this important safety consideration on page 12, line 399-419, contextualizing the accumulation data with our functional and biochemical safety markers.
References: National Toxicology Program (2010). NTP toxicology and carcinogenesis studies of chromium picolinate monohydrate (CAS No. 27882-76-4) in F344/N rats and B6C3F1 mice (feed studies). National Toxicology Program technical report series, (556), 1–194.
Anderson, R. A., Bryden, N. A., & Polansky, M. M. (1997). Lack of toxicity of chromium chloride and chromium picolinate in rats. Journal of the American College of Nutrition, 16(3), 273–279. https://doi.org/10.1080/07315724.1997.10718685
Comments 6: Considering that high doses of chromium picolinate can be toxic, could the authors explain how they determined that the doses used in this study were safe? Based on the recommendation.
Response 6: The dose of chromium picolinate (300 µg/kg BW) was carefully selected based on an extensive review of the scientific literature. This dosage is consistent with, or lower than, doses used in numerous previous rodent studies that have demonstrated metabolic benefits without any signs of toxicity which was in the range of 300 µg/kg BW (Zha et al., 2007; Majewski et al., 2022; Stępniowska et al., 2022) and, 1 and 10 mg/kg BW (Abdourahman & Edwards, 2008). Our own results, which showed no adverse effects on animal health or organ function, further confirm the safety of this dose in the context of our 8-week study. Moreover, According to the National Toxicology Program (2010), chromium picolinate monohydrate did not produce significant indications of toxicity in a 14-week study in rats and mice at dose levels up to 4,240 mg/kg bw/day for male rats (14133 times higher than our dose), 4,250 mg/kg bw/day for female rats, 11, 900 mg/kg bw/day for male mice and 9,140 mg/kg bw/day for female mice) (National Toxicology Program, 2010).
(National Toxicology Program, 2010)
We have added a sentence on page 15, line 571-574 in the methodology section (4.4) to explicitly state that the dose was chosen based on established literature demonstrating its safety and efficacy in rodent models, and we have included the relevant citations. References: Zha, L. Y.; Wang, M. Q.; Xu, Z. R.; Gu, L. Y., Efficacy of chromium(III) supplementation on growth, body composition, serum parameters, and tissue chromium in rats. Biol Trace Elem Res 2007, 119 (1), 42-50.
Majewski, M., Gromadziński, L., Cholewińska, E., Ognik, K., Fotschki, B., & Juśkiewicz, J. (2022). Dietary Effects of Chromium Picolinate and Chromium Nanoparticles in Wistar Rats Fed with a High-Fat, Low-Fiber Diet: The Role of Fat Normalization. Nutrients, 14(23), 5138. https://doi.org/10.3390/nu14235138
Stępniowska, A., Tutaj, K., Juśkiewicz, J., & Ognik, K. (2022). Effect of a high-fat diet and chromium on hormones level and Cr retention in rats. Journal of endocrinological investigation, 45(3), 527–535. https://doi.org/10.1007/s40618-021-01677-3
Abdourahman, A., & Edwards, J. (2008). Chromium Supplementation Improves Glucose Tolerance in Diabetic Goto-Kakizaki Rats. IUBMB life, 60, 541-548. doi:10.1002/iub.84
National Toxicology Program (2010). NTP toxicology and carcinogenesis studies of chromium picolinate monohydrate (CAS No. 27882-76-4) in F344/N rats and B6C3F1 mice (feed studies). National Toxicology Program technical report series, (556), 1–194.
Comments 7: The molecular weight determination of the isolated protein is presented, but details on the markers or standards used are missing.
Response 7: We sincerely apologize for this omission and thank the reviewer for pointing it out. The molecular weights were determined using two different pre-stained broad-range protein standards: Perfect Protein™ Markers (10-225 kDa) (M1) and ColorBurst™ Electrophoresis Marker (8-220 kDa) (M2), both purchased from Sigma-Aldrich (Darmstadt, Germany). The use of two distinct markers was a deliberate methodological choice to enhance the accuracy and reliability of our molecular weight estimations for the following reasons:
The specific molecular weights indicated on the gel image in Figure 1 (i.e., 216 kDa, 115 kDa, 96 kDa, 60 kDa, 52 kDa, 45 kDa, and 38 kDa) correspond to the bands from one of these standards and were used for the final size estimation. We have amended the methodology section (4.2.1) to include the names of the specific protein standards used on page 13, line 498-501.
Figure 1. SDS-PAGE analysis of sericin protein extracted from raw silk yarn. Extraction at a temperature of 121°C for durations of 60, 90, and 120 minutes is denoted by the symbols S60m, S90m, and S120m, respectively, with M1 and M2 representing the standard marker. Above is the new gel image with both markers and molecular weight. We have replaced it in the revised manuscript.
Comments 8: First paragraph of discussion should be added with the novelty of the study.
Response 8: This is an excellent and encouraging suggestion to improve the impact of our discussion. We agree that clearly stating the novelty upfront will better frame our findings for the reader. We have revised the beginning of the Discussion section on page 9-10, line 300-315. The new opening paragraph states that this is, to our knowledge, the first study to investigate silk sericin as a natural biopolymer to enhance the bioavailability and metabolic efficacy of chromium picolinate. It can highlight that our work provides novel mechanistic insight through FTIR analysis into a pH-dependent delivery system and demonstrates a synergistic effect on lipid and glucose metabolism, proposing a valuable new application for a sustainable byproduct of the silk industry.
Comments 9: Protein bands in the gel image are not clearly resolved.
Response 9: We acknowledge the reviewer's observation. This broad, somewhat smeared appearance is a well-documented and known characteristic of native silk sericin, and our results are consistent with those reported in other studies (Abdullah et al., 2025; Gupta et al., 2014).
(Abdullah et al., 2025)
(Gupta et al., 2014)
The diffuse nature of the bands is due to two main factors:
The purpose of our SDS-PAGE analysis was not to isolate a single, pure protein but rather to estimate the overall molecular weight range of the extracted mixture. This allowed us to confirm that the 60-minute extraction time yielded the most intact (highest molecular weight) sericin for use in our study, a goal which the gel successfully achieved. We have added a sentence to the explanation of Figure 1 to clarify that the broad nature of the bands is characteristic of the heterogeneous sericin protein mixture on page 3, line 129-130, citing appropriate literature to support this statement.
References: Abdullah, M., Xu, Z., Gao, S., Meng, K., & Zhao, H. (2025). Green Extraction and Separation of Silk Sericin with High and Low Molecular Weight and Their Gelling Performances. ChemistrySelect, 10. doi:10.1002/slct.202501940
Gupta, D., Agrawal, A., & Rangi, A. (2014). Extraction and characterization of silk sericin. Indian Journal of Fibre and Textile Research, 39, 364-372.
Comments 10: SS extracted at 121°C showed decreasing molecular weight over time: 60 min, 132–76 kDa (most intact); 90 min, 118–65 kDa; 120 min, 113–61 kDa.
Response 10: The reviewer's observation is correct and accurately summarizes the data. This progressive decrease in molecular weight is due to the thermal hydrolysis of the protein during the prolonged autoclaving process. This result was critical for our study design, as it confirmed that the 60-minute extraction was the optimal duration to obtain the most intact sericin. We hypothesized that this higher molecular weight protein would possess superior chelating properties compared to the more degraded fragments produced by longer extraction times.
Comments 11: Almost all the references should be revised based on MDPI guidelines.
Response 11: We sincerely apologize for this oversight. We have reviewed and reformatted the entire reference list to ensure it fully complies with the International Journal of Molecular Sciences' (MDPI) specific guidelines for authors before resubmission.
Below is the list of new references added in the revised manuscript:
We are confident that addressing these points will significantly strengthen our manuscript. We thank you again for your valuable time and expertise.
|
|
4. Response to Comments on the Quality of English Language |
|
Point 1: The English is fine and does not require any improvement. |
|
Response 1: We thank you for your positive comment on our work quality of English language. |
|
|
|
|
|
|

Round 2
Reviewer 1 Report
Comments and Suggestions for Authors
Thank you for the detailed responses and the revisions implemented in the manuscript. All of my concerns have been fully addressed, and I am satisfied with the current version of the work.